# Concerns about Household Violence during the COVID-19 Pandemic

**DOI:** 10.3390/ijerph192214633

**Published:** 2022-11-08

**Authors:** Dawn-Li Blair, Margot Shields, Lil Tonmyr

**Affiliations:** Public Health Agency of Canada, Ottawa, ON K1A 0K9, Canada

**Keywords:** coronavirus, family violence, household violence, child maltreatment, household size, household income

## Abstract

Evidence about how the pandemic affected household violence in Canada is mixed, but inarguably, the risk factors increased. This study used data from the 2020 Canadian Perspective Survey Series and the 2020 and 2021 Surveys of COVID-19 and Mental Health to examine the following: changes in the prevalence of concern about violence in individuals’ own homes during the pandemic; the characteristics of those who expressed concern; and the prevalence of concerns for specific household members. Among Canadians, the prevalence of concern about violence in individuals’ own homes decreased significantly between July and Fall 2020 (5.8% to 4.2%). Among women, the characteristics that were significantly associated with higher adjusted odds of concern about household violence included larger household size and lower household income. Lower education among women was associated with lower adjusted odds of concern. The associations with higher adjusted odds of concern among men included: being an immigrant, larger household size, and lower household income. From Fall 2020 to Spring 2021, the prevalence of concerns for oneself and for a child/children increased (1.7% to 2.5% and 1.0% to 2.5%, respectively), but concern for other adults in the household decreased (1.9% to 1.2%). Ongoing surveillance is needed to understand vulnerable populations’ exposure to household violence and to inform policies and programs.

## 1. Introduction

The COVID-19 pandemic raised alarm about potential increases in family violence globally and in Canada [1,2,3,4,5]. Social isolation, heavier workloads, financial insecurity, and unanticipated caregiving duties added stress to individuals’ lives. Although social distancing and quarantines are effective measures to mitigate the spread of infection, these restrictions have unintended consequences. Such measures foster social isolation, a salient predictor of domestic violence [6,7]. Restrictions also result in functional isolation, whereby those who experience, or are concerned about, family violence have limited access to support systems [6,8].

Furthermore, other risk factors for household violence have been exacerbated by the pandemic. Examples include major depressive disorder [9], job loss [10], and substance abuse [11] among others [12]. 

Evidence on how the pandemic has affected the prevalence of household violence is inconsistent. Many studies are of limited quality and are based on non-representative and/or small samples (often crowd-sourced). A 2022 systematic review of changes in domestic violence (including intimate-partner violence and violence against children) during the pandemic concluded that there was an increase in the prevalence of psychological/emotional and sexual violence and in the severity of all types of domestic violence [13]. A 2021 systematic review found an increase in domestic violence globally [14]. Another review that included studies from mid-2020 reported a general decline in reports of child maltreatment but suggested that more research was required to monitor the impact of the pandemic [15]. A fourth 2021 review stated that it is unknown whether rates of violence have changed [16]. 

Pre-pandemic nationally representative trend data on household violence in Canada are limited. Studies using national survey data have found evidence of a decline in childhood sexual abuse since 1990 [17,18] and a decrease in childhood physical abuse among those born between 1980 and 1999 (birth cohort) [19]. Statistics Canada, Canada’s national statistics agency, has also found a decline in self-reported spousal violence since 2004 [20].

Since the onset of the pandemic, Statistics Canada has reported that in a survey of victim services, 54% reported an increase in the number of domestic violence victims they served early in the pandemic [21,22].

During the pandemic, researchers have been hesitant to inquire directly about recent experiences of household violence, given possible confinement with, and dependence on, perpetrators. Statistics Canada developed a measure that approached the subject indirectly by asking respondents if they were concerned about violence in the home. This measure was included in two surveys between 29 March and 24 April 2020 (cross-sectional web-panel survey and crowd-sourcing survey) and yielded estimates of 8% to 11% of Canadians being “very” or “extremely” concerned about violence in the home [23,24,25]. In partnership with the Public Health Agency of Canada (PHAC), Statistics Canada subsequently modified the measure to ask about concern about violence in respondents’ own homes and to identify the targets of concern. These modifications provide: (1) specificity, by asking about concern regarding the respondents’ own homes (rather than general concern in the wider population), and (2) details about whom the respondents are concerned for in their homes. 

This analysis uses data from the 2020 Canadian Perspectives Survey Series and the 2020 and 2021 Surveys of COVID-19 and Mental Health to investigate: The prevalence among Canadians of concern about violence in their homes from July 2020 to May 2021.The sociodemographic characteristics associated with reporting concern.Information on which household members Canadians were concerned about being the targets of violence.

Such public health data are crucial for monitoring temporal population-level health changes and identifying inequitable exposure across sociodemographic groups. 

## 2. Methods

### 2.1. Data Sources

#### 2.1.1. Canadian Perspective Survey Series

The Canadian Perspectives Survey Series (CPSS) is an online web-panel survey. The target population is provincial residents aged 15 or older; however, to match the Survey of COVID-19 and Mental Health (see below), the analysis included only data from respondents 18 or older. Individuals living on reserves, in institutions, in the territories, or in extremely remote areas with low population density were excluded. Together, these exclusions represent fewer than 2% of the population aged 15 or older. The present analysis used data from CPSS cycle 4. Responses from this representative survey were collected from 20 to 26 July 2020. This cycle was sent to 7424 panelists (response rate = 58.2%; n = 4180). Additional information about this survey, including the questionnaire, can be found elsewhere [26].

#### 2.1.2. Survey of COVID-19 and Mental Health

The Survey of COVID-19 and Mental Health (SCMH) is a cross-sectional representative survey administered online or by telephone. The target population is residents of the provinces and territorial capitals aged 18 or older. Individuals living on reserves, in institutions, in collective dwellings, or outside the capital cities of the territories are excluded. Together, these exclusions represent fewer than 2% of the population. 

The present analysis used data from the 2020 and 2021 cycles of the SCMH. The 2020 SCMH was sent to a random sample of 30,000 dwellings (response rate = 53.3%); the 2021 SCMH was sent to 18,000 dwellings (response rate = 49.3%). For both cycles, dwellings were chosen from the Dwelling Universe File (a list of dwellings based on administrative data, created by Statistics Canada). From each occupied dwelling, one respondent was randomly chosen to complete the survey. Responses for 2020 were collected from 11 September to 4 December; 2021 data were collected from 1 February to 7 May. Respondents were asked for permission to share their data with PHAC; 12,344 respondents from the 2020 cycle and 6592 from the 2021 cycle granted permission. Additional information about this survey, including the questionnaire, can be found elsewhere [27].

### 2.2. Measures

#### 2.2.1. Concern about Violence

The 2020 and 2021 SCMH cycles and cycle 4 of the CPSS asked respondents if they were concerned about violence in their homes. 

The question in the SCMH cycles was: “How concerned are you about violence in your home?” Respondents selected one of the following: “Not at all,” “Somewhat,” “Very,” or “Extremely.” The question was accompanied by informational text designed to prepare respondents for potentially sensitive material and to inform them about mental-health resources. 

In the CPSS, “Concern about violence” was presented in a list of potential pandemic-related concerns. Respondents were asked: “How concerned are you about each of the following impacts of COVID-19?” The sub-question for ‘Concern about violence’ was “Violence in your home,” with the same response options as the SCMH. 

For each survey, responses to concern about violence in the home were dichotomized into “Yes” (“Somewhat,” “Very,” or “Extremely”) or “No” (“Not at all”). Although level of concern is important, this dichotomy recognizes that any concern about violence is alarming. Furthermore, grouping the responses provided an adequate sample size for analysis of subpopulations.

#### 2.2.2. Target of Violence

In the SCMH, respondents who reported concern about violence in their homes were asked a follow-up question: “Whom in your household are you concerned about being a target of violence?” Respondents could select all that applied from: “Yourself,” “Another adult or other adults in the household,” and “Child or children.” 

#### 2.2.3. Sociodemographic Characteristics

Concern about violence in individuals’ homes was examined across several sociodemographic variables: gender, age group, immigrant status, place of residence, household size, presence of a household member younger than 18, marital status, job status, highest level of educational attainment, household income quintile, and region. Marital status was not available from the 2020 SCMH. 

### 2.3. Analysis

Estimates of the prevalence of concern about violence in the home and 95% confidence intervals (CIs) were calculated for the total sample and by gender for each survey cycle. Analysis of gender-diverse individuals was not possible due to inadequate sample size. 

Since the overall prevalence of concern about violence in the home was stable between SCMH 2020 and 2021, and interactions with survey year were not significant, data from the two cycles were combined to maximize sample size when examining associations between sociodemographic characteristics and concern about violence. Given that both cycles of the SCMH used nearly identical methodologies, and their collection periods were relatively close in time, Statistics Canada deemed it acceptable to combine the datasets [28]. Adjusted odds ratios simultaneously controlling for all sociodemographic characteristics were calculated using logistic regression. CPSS data were not included because of differences in questionnaire items. Statistics Canada’s “release guidelines for quality” were applied to all estimates included herein [28].

Weights provided by Statistics Canada were used for all analyses to ensure that the data were representative of the population and to make adjustments for non-respondents. To account for survey-design effects, standard errors, coefficients of variation, and 95% CIs were estimated using the bootstrap method [29]. Significant differences between prevalence estimates (*p* < 0.05) were determined using chi-square tests. Analysis of differences between regions used Bonferroni adjustments for multiple comparisons. All analyses were conducted in SAS Enterprise Guide version 7.1 (SAS Institute Inc., Cary, NC, USA).

## 3. Results

### 3.1. Estimates over Time

Table 1 shows estimates from the 2020 CPSS and the 2020 and 2021 SCMH of the prevalence of Canadians’ concern about violence in their own homes (henceforth referred to as “concern about violence”). In July 2020, the overall prevalence was 5.8%, but by Fall 2020, it had dropped significantly, to 4.2%. The prevalence in Spring 2021 (5.0%) was not significantly different from either of the two 2020 figures. No significant changes over time were observed by gender. 

### 3.2. Sociodemographic Associations

When sociodemographic characteristics were examined in logistic regression models using combined 2020 and 2021 SCMH data, gender was not significantly associated with concern about violence (Table 2).

For women, being in a two-person or a three-or-more-person household (with or without a household member under 18) was associated with higher odds of expressing concern about household violence, compared to living alone (AOR of 1.9 to 3.9). Women with jobs had higher odds of reporting concern than those who had not had a job in the previous week (AOR = 1.8). Women in the three lowest household income quintiles had higher odds of being concerned about violence, compared with those in the highest quintile (AOR of 1.9 to 3.1). By contrast, having high school education or less was associated with lower adjusted odds, compared with having a university credential above the bachelor’s level (AOR = 0.5). Women in Atlantic Canada had significantly lower odds of reporting concern about household violence, compared with women in Ontario (AOR = 0.5). 

Among men, immigrants had higher odds of expressing concern about violence in their homes, compared with non-immigrants (AOR = 1.6). Higher odds were also observed for men living in two-person households without a household member under 18 and in three-or-more-person households (with or without a household member under 18) compared to living alone (AOR of 2.0 to 5.7). Men in the lowest household income quintile had higher odds of reporting concern, compared with those in the highest quintile (AOR = 2.5). Men living in Quebec had higher odds of reporting concern than did those in Ontario (AOR = 2.3). 

The significant associations for the total population largely corresponded to those found in the gender-stratified analysis.

### 3.3. Target of Violence

Table 3 provides the prevalence of concern about household violence, by target of violence, during the 2020 and 2021 SCMH. In Fall 2020, 1.9% of Canadians were concerned for another adult/adults in their home; 1.7% were concerned for themselves; and 1.0% were concerned for a child/children. In Spring 2021, a significantly lower percentage were concerned for another adult/adults (1.2%), but the percentages concerned for themselves (2.5%) or for a child/children (2.5%) rose significantly. In the gender-stratified analysis, only the increase in concern for children was significant for both genders (from 1.0% to 3.0% among females, and 1.0% to 2.1% among males). 

## 4. Discussion

The percentage of Canadians reporting concern about violence in their home decreased significantly from July to Fall 2020 (5.8% to 4.2%); the estimate for Spring 2021 (5.0%) was not significantly different from the two earlier figures. Among women, larger household size (with or without a household member under 18), having a job in the previous week, and household income in the three lowest quintiles were associated with higher adjusted odds of concern about household violence. Lower levels of education and living in Atlantic Canada were associated with lower adjusted odds. Among men, being an immigrant, larger household size, household income in the lowest quintile, and living in Quebec were associated with higher adjusted odds of concern. From Fall 2020 to Spring 2021, the prevalence of concerns for oneself and for a child/children increased, but concern for other adults in the household decreased. 

### 4.1. Strengths and Limitations 

The primary strength of this study is that it is based on large representative samples from the Canadian provinces and territorial capitals. The data were collected at three periods during the pandemic and, thus, offer a unique opportunity to investigate levels of concern about household violence during the pandemic. The analysis was conducted using several sociodemographic characteristics, including immigrant status, household size, and household income. Furthermore, SCMH data allowed for more specific identification of targets of violence. 

A limitation is the exclusion of certain subpopulations (for example, individuals younger than 18 and residents of remote areas, reserves, and collective dwellings), among whom the prevalence of concern about violence may differ from the estimates reported in this study. Additionally, differences between the CPSS and SCMH methods may have affected the results.

#### Measuring Concern about Violence

This study must be considered in the context of the pandemic, which confined individuals to their homes with potential perpetrators of violence and limited access to social support and services. Ensuring the safety and health of Canadians was a crucial priority in developing a method for measuring family violence that precluded the inclusion of behaviour-based violence questions with established validity and reliability. More specifically, direct questions regarding experiences of and exposure to household violence could not be used as they may have endangered or distressed respondents. Therefore, a new measure was used to instead explore concern about violence in the home. Although it is not equivalent to actual experiences of household violence (and cannot be interpreted as such), it is an innovative measure that allows insights into household violence during the unprecedented pandemic. 

A limitation of this measure is the lack of comparable pre-pandemic data; it is unknown whether the prevalence rates of concern about household violence reported here differ significantly from those in previous years. Furthermore, because the data were collected at three different times during the year, the results may have been affected by unknown seasonality bias. Data were available about the potential targets of violence, but not about its perpetrators. This raises questions about whether the respondents were concerned about violence perpetrated by themselves, another household member, or someone external to the household (for example, an ex-partner). Whether the respondents were concerned about being a perpetrator or victim of violence (or both) could have had an unknown effect on the estimates reported herein [30,31,32,33]. A further limitation is that the estimates of concern about violence over time and sociodemographic associations relied on the primary measure despite the indication of differences in concern by target of violence over time. The primary measure disregards these differences, but a detailed analysis by target of violence was not possible due to the small cell size. Lastly, the data were self-reported, and the extent to which concern about violence may have been underestimated because of respondents’ reluctance or inability to disclose sensitive information is unknown. This is also a limitation of other types of self-reported data on family violence; false negatives are likely to be common, although false positives are rare [34].

Future investigations of the validity and reliability of concern about violence measures as approximate measures of experiences of violence, and as possible indicators of increased risk of household violence, would be valuable. It is possible that if household violence occurs, respondents may be more likely to acknowledge that they have concerns but be less likely to divulge that violence has occurred. Furthermore, if concern about violence is expressed, this may have clinical relevance, even if violence has not occurred. When measuring the validity of the concern about violence, it may be important to consider respondents’ characteristics, including gender, socioeconomic status, and whether the respondent is a potential perpetrator or victim of violence (or both). Additionally, future research may investigate associations between concern about violence with non-sociodemographic variables. 

Consideration should also be given to the reference period used in this study, which was unspecified. Since no time frame was given, the implication was that the respondents should reply according to how they felt in the moment, with minimal time in which to reflect. This lack of timeframe may have failed to capture respondents who had previously been concerned about violence, but were no longer concerned when they responded.

### 4.2. Interpretation

The high prevalence of concern about violence in individuals’ own homes in July 2020 corresponds to an increase in police-reported family violence [35], indicating that concern about household violence could be a reasonable proxy measure for experiences of household violence. The decrease in concern from July to Fall 2020 may have been related to the easing of public-health restrictions (for example, most public schools had re-opened for in-person learning [36]), the implementation of new practices by service providers [37,38], and increased social cohesion [39]. Additionally, intense media attention early in the pandemic may have increased concern, which could have subsided by Fall 2020.

The associations with household size and the presence of a household member younger than 18 indicate that as household size increases, so does concern about violence. This may reflect fears caused by the presence of more potential perpetrators and targets of violence. Moreover, members of larger households may have experienced more stress during the pandemic [40,41], which could have translated into increased concern about household violence. Resources for preventing household violence should perhaps target larger households.

Single parents are likely to constitute a substantial share of individuals living in two-person households with a member younger than 18. This household composition, compared to living alone, was associated with significantly higher levels of concern about household violence among women, but not among men. Higher levels of concern are possibly related to previous experiences of violence among single mothers, which single fathers are less likely to experience [42,43,44]. Alternatively, the context and stress related to being a single mother may make them more likely to have concern about violence [45] either perpetrated by themselves or by someone external to the household.

Women with high school education or less were significantly less likely to report concern about violence in their home than those with a university education above the bachelor’s level. By contrast, previous studies found associations between low education and an increased risk of violence [46,47]. A possible explanation for this is that women with higher education may have had more awareness of potential household violence and may have consumed more relevant media [48], which could have increased their concern about violence early in the pandemic. However, the association between lower household income and greater concern about violence is consistent with earlier research on the associations between income and family violence [46,47,49,50,51]. 

The increase over time in concerns for oneself or for a child/children may reflect the compounding effects of prolonged restrictions and public health measures. During Spring 2021, the number of COVID-19 cases increased, and new restrictions were introduced [36]. However, this does not explain the decrease in concern for other adults; a possible explanation for this is that individuals are less concerned about perpetrating violence towards other adults in their home. The higher levels of concern about violence against children are particularly unsettling as experiences of violence during childhood can have severe and long-lasting effects on a child’s physical and mental health [52].

## 5. Conclusions

This analysis provided data on Canadians’ concern about household violence at three different times during a public health emergency. Sociodemographic associations with such concern were identified, as were targets of concern. Measuring concern about violence is an alternative method for gaining information about household violence; however, research is warranted to determine the validity and reliability of this measure. Beyond exploring concern about violence, a consistent and robust system for monitoring and researching different types of household violence that includes multiple information sources is needed to more fully understand how the pandemic has affected and continues to affect vulnerable populations’ exposure to household violence. This information, in turn, could inform prevention policies and victim-support programs. In the meantime, the current study and its measures of concern about household violence can be used to provide an indication of where resources should be allocated.

## Figures and Tables

**Table 1 ijerph-19-14633-t001:** Prevalence of concern about violence in individuals’ own homes, by gender, population aged 18 or older, Canada, 2020 and 2021.

	CPSS Cycle 4		SCMH Cycle 1		SCMH Cycle 2	
	July 2020		Fall 2020		Spring 2021	
	%	95% CI	%	95% CI	%	95% CI
Total population, 18+	5.8	(4.6, 7.1)	4.2 *	(3.6, 4.9)	5.0	(4.2, 6.0)
Gender						
Female	5.2	(3.8, 6.9)	3.9	(3.2, 4.8)	5.0	(3.9, 6.4)
Male	6.3	(4.5, 8.5)	4.5	(3.6, 5.6)	5.0	(3.8, 6.5)

Sources: 2020–2021 Survey on COVID-19 and Mental Health (SCMH); 2020 Canadian Perspective Survey Series (CPSS). Abbreviations: CI = Confidence Interval. * Significantly different from CPSS (*p* < 0.05).

**Table 2 ijerph-19-14633-t002:** Adjusted odds ratios for concern about violence in individuals’ own homes, by gender and sociodemographic characteristics, population aged 18 or older, Canada, 2020/2021.

	SCMH Cycle 1 and Cycle 2 Combined					
	Total		Female		Male	
	AOR	95% CI	AOR	95% CI	AOR	95% CI
Gender (reference is male)						
Female	1.0	(0.8, 1.3)				
Age group (reference is 65 or older)						
18–34	1.0	(0.6, 1.9)	0.8	(0.3, 1.9)	1.2	(0.6, 2.5)
35–49	1.4	(0.8, 2.6)	1.0	(0.4, 2.4)	2.0	(1.0, 3.9)
50–64	1.3	(0.8, 2.2)	0.9	(0.4, 1.9)	1.9	(1.0, 3.5)
Immigrant status (reference is non-immigrant)						
Immigrant	1.5 *	(1.1, 2.0)	1.4	(1.0, 2.1)	1.6 *	(1.1, 2.5)
Place of residence (reference is rural)						
Urban	0.8	(0.6, 1.2)	0.9	(0.6, 1.4)	0.8	(0.4, 1.4)
Household size and presence of household member younger than 18 (reference is 1)						
2, with household member under 18	2.6 *	(1.1, 6.3)	3.3 *	(1.2, 9.4)	0.3	(0.0, 42.3)
2, no household member under 18	1.9 *	(1.3, 2.8)	1.9 *	(1.1, 3.3)	2.0 *	(1.1, 3.8)
3+, with household member under 18	4.3 *	(2.8, 6.6)	3.9 *	(2.1, 7.2)	5.3 *	(2.8, 10.0)
3+, no household member under 18	4.6 *	(2.8, 7.3)	3.7 *	(1.8, 7.4)	5.7 *	(2.9, 11.2)
Marital status (reference is single never married; SCMH 2021 only)						
Married/Common-law	0.7	(0.4, 1.3)	0.8	(0.4, 1.8)	0.7	(0.3, 1.7)
Separated/Divorced	0.7	(0.3, 1.5)	0.6	(0.2, 1.5)	0.9	(0.2, 5.2)
Job status (reference is did not have a job in previous week)						
Had a job in previous week	1.2	(0.8, 1.7)	1.8 *	(1.0, 3.0)	0.8	(0.5, 1.4)
Highest level of educational attainment (reference is postsecondary certificate, diploma or degree)						
High school or less	0.8	(0.5, 1.2)	0.5 *	(0.3, 1.0)	0.9	(0.5, 1.7)
Postsecondary certificate, diploma or degree	0.9	(0.6, 1.3)	1.1	(0.7, 1.7)	0.7	(0.4, 1.2)
Household income quintile (reference is 5)						
1 (lowest)	2.8 *	(1.7, 4.7)	3.1 *	(1.6, 6.1)	2.5 *	(1.1, 5.6)
2	2.1 *	(1.3, 3.3)	2.0 *	(1.1, 3.7)	2.0	(0.9, 4.3)
3	1.9 *	(1.2, 3.0)	1.9 *	(1.0, 3.4)	1.8	(0.8, 3.8)
4	1.1	(0.7, 1.9)	0.9	(0.5, 1.8)	1.3	(0.6, 3.0)
Region (reference is Ontario)						
Atlantic	0.8	(0.5, 1.3)	0.5 *	(0.3, 0.9)	1.1	(0.6, 2.5)
Quebec	1.6 *	(1.0, 2.5)	1.1	(0.6, 2.0)	2.3 *	(1.2, 4.7)
Prairies	1.0	(0.6, 1.7)	0.7	(0.4, 1.3)	1.5	(0.7, 3.3)
British Columbia	1.6	(0.9, 2.7)	1.2	(0.6, 2.5)	2.0	(0.8, 4.6)
Territorial capitals	1.6	(0.9, 2.7)	1.5	(0.7, 3.0)	1.6	(0.7, 3.5)

Sources: 2020–2021 Survey on COVID-19 and Mental Health (SCMH). Abbreviations: AOR = Adjusted Odds Ratio. CI = Confidence Interval. Note: A Bonferroni adjustment for multiple comparisons was made when comparing estimates for regions; 99% CIs are presented. * Significantly difference from reference (*p* < 0.05; *p* < 0.01 for region).

**Table 3 ijerph-19-14633-t003:** Prevalence of concern about violence in individuals’ own homes, by gender and target of violence, population aged 18 or older, Canada, 2020 and 2021.

	SCMH Cycle 1		SCMH Cycle 2	
	Fall 2020		Spring 2021	
Target of Violence	%	95% CI	%	95% CI
Total population				
Self	1.7	(1.3, 2.1)	2.5 *	(1.9, 3.2)
Other adult(s)	1.9	(1.5, 2.4)	1.2 *	(0.8, 1.7)
Child or children	1.0	(0.7, 1.3)	2.5 *	(1.9, 3.3)
Female				
Self	1.8	(1.3, 2.4)	2.7	(1.9, 3.7)
Other adult(s)	1.6	(1.1, 2.2)	1.0	(0.6, 1.6)
Child or children	1.0	(0.6, 1.5)	3.0 *	(2.0, 4.3)
Male				
Self	1.6	(1.1, 2.2)	2.3	(1.4, 3.5)
Other adult(s)	2.2	(1.5, 3.1)	1.4	(0.8, 2.2)
Child or children	1.0	(0.6, 1.5)	2.1 *	(1.3, 3.1)

Sources: 2020–2021 Survey on COVID-19 and Mental Health (SCMH). Abbreviations: CI = Confidence Interval. * Significantly different from SCMH Cycle 1 (*p* < 0.05).

## Data Availability

Restrictions apply to the availability of these data. Data were obtained from Statistics Canada’s Virtual Data Lab and are available with the permission of Statistics Canada.

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
