# Peer review of "Concerns about Household Violence during the COVID-19 Pandemic"

_ijerph, 2022, doi:10.3390/ijerph192214633_

Round 1

Reviewer 1 Report (Previous Reviewer 2)

Concerns about domestic violence during the COVID-19 pandemic.

The authors tried to respond to the comments and suggestions from the review. They agreed with most of the comments. However, with regard to some of the comments, they did not think they could use them to modify the text.

I am wondering about the attitude / interpretation of the answer to the question about the increased anxiety and fear related to the possibility of violence in the single parent's family. The issue of parents' loneliness - there has been an increase in anxiety about own safety (and / or the child's safety) in the family. Of course, it is possible that such violence will happen - the question is: who will be the perpetrator? absent spouse or other family member - someone who is in the same household? Or maybe the respondent will be the perpetrator himself or his child?

Author Response

This manuscript is a resubmission of an earlier submission. The following is a list of the peer review reports and author responses from that submission.

Round 1

Reviewer 1 Report

Dear authors, thank you very much for the opportunity to get acquainted with the content of your text, which deals with a very topical and sensitive issue. I believe the topic is so important that it deserves a more detailed analysis and a broader discussion.
I, therefore, recommend that the text be elaborated as follows:
There is a complete lack of literature review. This is an issue that is widely described in the literature. A more in-depth reflection on current knowledge would therefore be helpful.
In the Methods chapter, the information is repetitive - lines 56-57 x 66-67 - and the data collection dates are unclear. Please also describe in more detail the composition of the research population. How did your hypotheses sound?
The results of the statistical analyses are insufficiently commented. Please do not leave it to the reader to extract information from the tables presented. As authors of the text, you have to highlight any significant results you find.
The discussion is also inadequate, which is undoubtedly related to the introduction's lack of a literature review.

Reviewer 2 Report

Undoubtedly, the issue of domestic violence is an important topic - regardless of whether we study it in times of pandemics, world wars, economic crises and other negative socio-historical phenomena or ecological catastrophes.

The image obtained by the authors - in my opinion - seems to be moderately disturbing, perhaps due to the limitations in the implementation of analyzes, for which researchers: including ethical considerations or psychological comfort of the respondents (and probably concern for the well-being of the groups included in the research: crowd-sourcing survey). Is it correct to assume that by asking indirect questions about domestic violence, it will be possible to secure potential victims against the perpetrator of violence (in an anonymous online survey)? Another explanation can be considered: not all people experiencing domestic violence - want to reveal themselves as victims. Besides, it may already be such a degree of family entanglement and dependence on the perpetrator and the violence itself that the victim does not perceive this / his/her situation as pathological. Another explanation is also possible, namely in a situation of a serious threat to health and life due to a disease for which during the exploration period (and still) there is no cure - domestic violence somehow receded into the "background", as - in the opinion of the victims - less dangerous from the virus.

The authors focus primarily on quantitative analyzes and sociodemographic data. Not all readers are aware of how data (item content, questions?) Is collected by the Canadian Perspective Survey Series and the 2020 and 2021 Surveys of COVID-19 and Mental Health.

The authors rightly point to the inability to refer to pre-Covid 19 data. However, would it be possible to present data on reported domestic violence during the pandemic period and compare it with the number of reports in the pre-pandemic period? For this purpose, it would be advisable to use the court, police, social (and other) statistics from the regions described in this article. Of course, it cannot be certain that all domestic violence incidents have been reported during the pandemic period - as in previous periods. Unfortunately, domestic violence is still largely hidden (somewhat in the shadows).

Three goals have been achieved. However, when it comes to goal no. 3, it seems that the answers obtained confirmed the social knowledge: an increase in anxiety about the child and then about one's own person and the tendency to protect the offspring.

In view of the events in 2022 - new strains of the virus and new socio-economic and ecological threats, it is justified to continue exploration - maybe it is worth considering expanding the field of variables, refining tools and selecting new ones.

Round 2

Reviewer 1 Report

Dear authors, I am very sorry, but I have to stand by my opinion from the previous review. Strong theoretical foundations and the resulting interpretations are the basis of good scientific work. I agree that “much of this information, especially from early in the pandemic, was of limited quality and was based on non-representative and/or small samples.” However, even this published research forms an integral part of the academic discussion. I would expect new studies to come to terms with these research limitations. In my opinion, this is how science works – it evolves, and we should reflect on its evolution. Suppose you believe that there is no quality research before your research. In that case, you need to specify its limitations both in the introduction and in the discussion and write why your (much higher quality) research arrived at different results. And if you did not arrive at different results, what about that quality?
